# Gauging away the Stoner model:
# Engineering unconventional metallic ferromagnetism with artificial gauge fields

Joshuah T. Heath* [1], Kevin S. Bedell[1],

[**1**] Physics Department, Boston College, Chestnut Hill, Massachusetts 02467, USA
heathjo@bc.edu

October 23, 2020

## Abstract

**For nearly a century, the Stoner model has dominated research in itinerant ferromagnetism, yet recent work on ultracold, optically trapped Fermi gases suggests that phase separation of spins occurs independent of long-range magnetic order. In this paper, we consider the breakdown of the Stoner criterion in a Landau-Fermi liquid with a weak non-zero gauge field. Due to a process analogous to Kohn's theory of a metal-insulator transition, we find the stability of a Fermi liquid phase in the absence of non-quasiparticle contributions to the response is strongly dependent on Landau parameters of mixed partial waves. Our work paves the way for the description of novel spintronic hardware in the language of Landau-Fermi liquid transport.**

---

The study of artificial gauge fields has recently gained traction as a means of exploring physics beyond conventional electromagnetism [1–4]. The creation of a spin-dependent gauge field through synthetic spin-orbit (SO) coupling is of particular interest in the field of spintronics as a means to create a finite spin current [1,5–11], facilitating the development of novel solid state devices which exploit the electron's spin degrees of freedom. As we approach the inevitable termination of Moore's law, such machinery (and, subsequently, the gauge fields which support them) will prove to be indispensable for the development and implementation of new information storage and logic devices.

In addition to a spin-dependent gauge field, the generation of spin polarization necessary for spintronic hardware usually requires the presence of a non-equilibrium spin bath, by which spin-polarized transport can be studied [12] and relaxation back to equilibrium measured and ultimately controlled for the efficient implementation of fast switching [13,14]. The itinerant ferromagnetism required for such non-equilibrium behavior was originally described in terms of the Stoner paradigm [15, 16], in which a repulsive exchange interaction in the spin-1/2 Fermi gas results in a segregation of delocalized spins into distinct domains [17–19]. Nevertheless, a significant number of theoretical [20–23] and experimental [24, 25] works have suggested that quantum correlational effects neglected in Stoner's original mean-field argument competes with ferromagnetism. Similar work on ultracold gases of $^6$Li atoms suggests that the phenomenon of spin segregation occurs independently of ferromagnetic ordering [26–28], once again in violation of Stoner's model. As the former point contradicts the theory of ferromagnetic Fermi liquids [29–35] and the latter contradicts the standard Pomeranchuk argument, the applicability of Landau-Fermi liquid theory to materials of spintronic interest is called into question if Stoner's argument

remains problematic, threatening to backtrack significant progress made in the marriage of the two fields [12, 36–41].

In this paper, we show that a stable metallic phase can coexist with ferromagnetic order in the absence of spin segregation as long as the system supports some finite, weak gauge field. A non-zero vector potential results in a universal translation of the entire Fermi sea, and subsequently induces a novel interdependence between Landau parameters of unequal partial waves if we are to ensure adiabatic continuity with the non-interacting system remains valid. By taking into account dynamically-generated instabilities of zero sound, we find that the metallic phase for non-zero gauge field is protected by phase separation irrespective of long-range ferromagnetic ordering. This differs greatly from the traditional Stoner model, where phase separation and magnetic order occur synchronously. In this way, we show that something as simple as a gauge field can induce non-trivial momentum dependence in the near-equilibrium degrees of freedom which stabilize the Fermi liquid ansatz against time-reversal symmetry breaking.

Fermi surface instabilities of a standard Landau-Fermi liquid occur when the free energy functional

$$\delta \mathcal{F} \approx \sum_{p\sigma}(\epsilon_{p\sigma} - \mu)\delta n_{p\sigma} + \frac{1}{2}\sum_{pp'\sigma\sigma'}f_{pp'}(\sigma, \sigma')\delta n_{p\sigma}\delta n_{p'\sigma'} \tag{1}$$

becomes negative, where $\epsilon_{p\sigma}$ is the quasiparticle energy, $\delta n_{p\sigma} = n_{p\sigma} - n_{p\sigma}^0$ is the displacement of the distribution function, and the Landau parameter $f_{pp'}(\sigma, \sigma')$ is the second functional derivative of the total energy with respect to $\delta n_{p\sigma}$. Expanding the Landau parameters in terms of spherical harmonics, we can recast the above as

$$\delta \mathcal{F} = \frac{N(0)}{8}\sum_{\ell}\frac{1}{2\ell+1}\left[\nu_{\ell s}^2\left(1 + \frac{F_\ell^s}{2\ell+1}\right) + \nu_{\ell a}^2\left(1 + \frac{F_\ell^a}{2\ell+1}\right)\right] \tag{2}$$

where $f_{pp'}(\sigma, \sigma') = \frac{1}{N(0)}\sum_\ell\left(F_\ell^s + \sigma\cdot\sigma'F_\ell^a\right)P_\ell(\cos\theta)$, $N(0)$ is the density of states, and the Fermi surface distortion $\nu_{\ell\,s/a} = \sum_\ell\left(\nu_{\ell\uparrow}\pm\nu_{\ell\downarrow}\right)P_\ell(\cos\theta)$ is defined by $\delta n_p = -\frac{\partial n_k^0}{\partial\epsilon_k}\nu_p$.

In the s-wave channel, the stability condition $\delta\mathcal{F} > 0$ rests solely on the values of the symmetric ($F_0^s$) and anti-symmetric ($F_0^a$) $\ell = 0$ Landau parameters. Because the condition $1 + F_0^s < 0$ ($1 + F_0^a < 0$) leads to a negative compressibility (spin susceptibility), violations of the Pomeranchuk condition in the (anti-)symmetric Landau parameter mark the onset of a structural phase transition mediated by long-wavelength charge density (spin) fluctuations. However, note that a violation of the stability condition in *one* of the Landau parameters does not necessarily imply a complete breakdown of the Fermi liquid ansatz. In the limit of weak (i.e., local) ferromagnetism, the symmetric and anti-symmetric Landau parameters are related by $F_0^s = -F_0^a/(1 + 2F_0^a)$ [42, 43], and only then (for general non-zero $\nu_{0s}$ and $\nu_{0a}$) does the resultant ferromagnetic instability occur concurrently with the segregation of spin populations at the Fermi surface. It therefore becomes apparent that a Landau-Fermi description of the itinerant ferromagnetism demonstrated in ultracold gases of $^6$Li atoms [27, 28] must be realized as an "unconventional" variant of ferromagnetism characterized by higher non-vanishing orbital partial waves [37].

Higher-order $\ell > 0$ Pomeranchuk instabilities have been the subject of recent interest, with previous studies suggesting that such instabilities in the spin channel are accompanied by a dynamically-generated, non-relativistic spin-orbit (SO) coupling [36, 37]. The existence of an "emergent" SO coupling from low-energy collective phenomenon was brought into question by [44], where the divergence of the quasiparticle susceptibility $\chi(\mathbf{q}, \omega)$ in the $\ell = 1$ channel is removed by virtue of a vanishing vertex coupling between Landau

quasiparticles and bare electrons [45]; i.e., the non-quasiparticle contribution to the susceptibility exactly cancels the divergence in the quasiparticle contribution. As a result, the full disappearance of an $\ell = 1$ instability can only be realized for a specific **k**-dependence of the form factor $\lambda_{\ell=1}^{s/a}(\mathbf{k})$ [46]. A generic order parameter with different functional dependence of $\lambda_{\ell=1}^{s/a}(\mathbf{k})$ will result in the onset of Pomeranchuk instabilities for $\ell > 0$, while Pomeranchuk instabilities towards phases with spontaneously generated charge/spin currents prove to be impossible [47, 48]. To remain consistent with the recent literature, we limit our study to those systems where the order parameter of the resultant phase has the same symmetry as the charge/spin current. This corresponds to a nearly isotropic system where the high-energy and low-energy contributions to the susceptibility do not enjoy a direct correlation [46].

The dynamic SO coupling described by [36, 37] can be reformulated in terms of an SU(2) gauge field [1, 49–53]. The effects of a gauge potential $\mathbf{A}_\sigma$ on the Fermi surface has been noted previously in connection with Kohn's theorem [54, 55] and the anomalous Hall effect [56], yet a detailed study of the resultant Pomeranchuk instability has not been explored. To address this issue, we consider a general vector potential $\mathbf{A}_\sigma$, where $\mathbf{A}_\uparrow = \mathbf{A}_\downarrow \equiv \mathbf{A}$ for a symmetric (i.e., electromagnetic-like or spin-independent) gauge field and $\mathbf{A}_\uparrow = -\mathbf{A}_\downarrow \equiv \mathbf{A}$ for an antisymmetric (i.e., Rashba/Dresselhaus-like [57–62] or spin-dependent) gauge. Note that in the presence of the many-body SO coupling which generates a spin-dependent gauge field, the Landau parameter takes on a more complicated form as a generalized tensor under rotations of spatial momenta [63]. In addition to terms proportional to $\delta_{\alpha\overline{\alpha}}\delta_{\alpha'\overline{\alpha}'}$ and $(\sigma_i)_{\alpha\overline{\alpha}}(\sigma_j)_{\alpha'\overline{\alpha}'}$ (where the subscripts denote matrix elements and $\sigma_i$ is the Pauli matrix), the presence of a net spin polarization and finite spin-orbit coupling implies terms proportional to $(\sigma_i)_{\alpha\overline{\alpha}}\delta_{\alpha'\overline{\alpha}'}$ and $\delta_{\alpha\overline{\alpha}}(\sigma_i)_{\alpha'\overline{\alpha}'}$ in the Landau interaction parameter, with generic coefficients proportional to the magnitude of the momenta **p** and **p**′. This is a direct consequence of a net spin polarization of all electrons in the Fermi sea and, hence, a net displacement of the Fermi seas of up- and down-spin particles in opposite directions, resulting in the violation of spherical symmetry of the Fermi surface and intrinsic dependence of $f_{pp'}(\sigma, \sigma')$ on the azimuthal quantum number $m$. For this reason, we restrict ourselves to the limit $\frac{e}{c}\mathbf{A}_\sigma << \mathbf{p}_F$, allowing us to take the previously mentioned scalar decomposition of the Landau parameter in terms of $F_\ell^s$ and $F_\ell^a$. Such an approximation has similarly been taken in the chiral Fermi liquid description of electron systems with weak Rashba [40] and Dresselhaus [64] SO coupling, 2D electron gases with zero external magnetic field [65], and weakly polarized Fermi liquids in a quasi-equilibrium state [66].

For some weak non-zero gauge field, the momentum of every particle in the Fermi sea is shifted by the same amount: $\mathbf{p}_F \to \mathbf{p}_F - \mathbf{A}_\sigma$ [55, 56], taking units where $e = c = 1$. The resultant Fermi surface distortion can be found by expanding the Heaviside step function about $p_F(\theta, \phi) = p_F - A_\sigma$:

$$\delta n_{p-A_\sigma,\sigma} = \Theta(p_F(\theta, \phi) - p + A_\sigma) - \Theta(p_F - p)$$

$$= \delta n_{p\sigma} - A_\sigma \delta(p_F - p) - A_\sigma \delta p_F \frac{\partial}{\partial p}\delta(p_F - p)$$

$$- \frac{1}{2}A_\sigma^2 \frac{\partial}{\partial p}\delta(p_F - p) \tag{3}$$

where $\delta p_F \equiv p_F(\theta, \phi) - p_F$ defines the anisotropic shift of the Fermi momentum at finite interaction. The first term $\delta n_{p\sigma} = -\delta p_F \delta(p_F - p) - \frac{1}{2}\delta p_F^2 \frac{\partial}{\partial p}\delta(p_F - p)$ is identical to what is seen in a traditional Fermi liquid, while the last three terms are specific to Fermi liquids with some finite gauge. After a tedious but straightforward calculation, we can expand

the first expression in Eqn. (1) for $\mathbf{A}_\sigma \neq 0$ in terms of partial waves, yielding

$$\sum_{p\sigma}(\epsilon_{p-A_\sigma,\sigma} - \mu)\delta n_{p-A_\sigma,\sigma} \approx \frac{N(0)}{8}\left[\sum_\ell \frac{1}{2\ell+1}\left\{(\nu_{\ell s}^2 + \nu_{\ell a}^2) + \frac{16A}{p_F}\epsilon_F\nu_{\ell,s/a}\right\} + \frac{48A^2}{p_F^2}\epsilon_F^2\right]$$
(4)

where $\nu_{\ell,s/a} = \nu_\uparrow \pm \nu_\downarrow$ for the symmetric and anti-symmetric gauges, respectively. In the above derivation, we assume that $\epsilon_F >> \nu_{s/a}$ to keep the expression tractable, and have set $|\mathbf{A}_\sigma| \equiv A$.

A similar but somewhat more complex calculation can be performed on the second term in Eqn. (1) for finite gauge field (see Supplemental Material). When the dust settles, we are left with

$$\sum_{pp'\sigma\sigma'} f_{pp'\sigma\sigma'}\delta n_{p-A_\sigma,\sigma}\delta n_{p'-A_{\sigma'},\sigma'}$$

$$\approx \frac{N(0)}{8}\left[\sum_\ell \frac{1}{2\ell+1}\left(\frac{1}{2\ell+1}\{F_\ell^s\nu_{\ell s}^2 + F_\ell^a\nu_{\ell a}^2\} + \frac{8A}{p_F}\frac{F_\ell^s}{2\ell+1}\epsilon_F\nu_{\ell,s/a}\delta_{\ell 0}\right) + \frac{64A^2}{9p_F^2}F_1^{s/a}\epsilon_F^2\right]$$
(5)

where $F_\ell^{s/a}$ is $F_\ell^s$ for the symmetric gauge field and $F_\ell^a$ for the anti-symmetric gauge. Note that the second term is proportional to $F_\ell^s$ independent of what variety of gauge field we consider, and only remains finite for the $\ell = 0$ channel.

We can see from the above that the presence of a non-zero gauge field refashions the original $\ell = 0$ Pomeranchuk instability into a generalized condition between $F_1^{s/a}$ and $F_0^{s/a}$. This is a direct consequence of the Landau-Fermi free energy transforming into a second-order polynomial with respect to the Fermi surface distortion $\nu_\ell$, as opposed to simply being dependent on $\nu_\ell^2$ as seen in a conventional Fermi liquid. The presence of a linear $\nu_\ell$ and constant term from the gauge field puts new constraints on an equilibrium value of $\nu_\ell$ to ensure that such distortions are real. This yields the following stability conditions for the spin symmetric and spin anti-symmetric gauge fields, assuming $F_0^{s/a} < -1$:

$$\mathbf{A}_\uparrow = \mathbf{A}_\downarrow, \quad F_0^s < -1 \rightarrow F_1^s \geq \frac{9}{4}\left(\frac{1}{1+F_0^s} + F_0^s\right)$$
(6a)

$$\mathbf{A}_\uparrow = -\mathbf{A}_\downarrow, \quad F_0^a < -1 \rightarrow F_1^a \geq \frac{9}{4}\left(\frac{1 - 3F_0^a + F_0^s(4 + F_0^s)}{1 + F_0^a}\right)$$
(6b)

where we assume that $\nu_{0a} = 0$ for the symmetric gauge field and $\nu_{0s} = 0$ for the anti-symmetric gauge. If we instead take $\nu_{0s} = 0$ ($\nu_{0a} = 0$) for the (anti-)symmetric gauge, Eqns. (6a) and (6b) are simplified to $F_1^{s/a} \geq -27/4$. Whereas the $\ell = 0$ stability condition for $\mathbf{A}_\sigma = 0$ is solely determined by $F_0^{s/a}$, the presence of some weak finite gauge field implies that the stability of the Fermi surface has strong dependence on higher-order Landau parameters. As the spin susceptibility is calculated from the change in the *local* energy of a quasiparticle as the result of spin displacement [67], the value of the spin susceptibility is not effected by a uniform translation of the Fermi sea and is thus a gauge-invariant quantity. This suggests that ferromagnetic ordering still occurs when $F_0^a < -1$ for $\mathbf{A}_\sigma \neq 0$, albeit "disentangled" from the phase separation that plagues the local model.

In Fig. 1, we can see under what conditions the Fermi liquid is stable for the symmetric gauge. The left-hatched region corresponds to the only combination of Landau parameters

$F_0^s$ and $F_1^s$ where equilibrium Fermi liquid theory completely breaks down in the presence of a spin-symmetric gauge. The right-hatched region corresponds to a dynamical instability; i.e., where there exists some critical $\nu_{0s} \neq 0$ that pushes $\delta\mathcal{F} < 0$. Such instabilities have previously been noted in ferromagnetic Fermi liquids with spin-orbit coupling, where the collective mode can drive the system to a Lifshitz transition [68]. Only in the white region above the top blue line of Fig. 1 is the Fermi liquid phase completely free from instabilities for all values of $\nu_{\ell s}$. In Fig. 2, we see where the Fermi liquid is stable in the anti-symmetric gauge, with the regime of phase separation for $\nu_{0s} \sim 0$ reduced to a small sliver near $F_0^s \sim -2$ as $F_0^a \to -1$ from below. Here, the right-hatched region corresponds to those $F_1^a$ values where a dynamical instability is possible for $F_0^a \in [-2, 0]$, while the left-hatched corresponds to where $\delta\mathcal{F} < 0$ indefinitely in this range. While both the symmetric and anti-symmetric gauge fields increase the stability of the Fermi surface, the latter nearly eliminates the threat of phase separation at near-equilibrium if we are in close proximity to a ferromagnetic instability.

Physically, we associate the newfound stability of the itinerant metallic phase with the condensation of gauge bosons. This conclusion is hinted at by repeating the calculation of Eqns. (4)+(5) for the $\ell = 1$ channel, in which we find the constraint $\frac{3}{8}\left(-13 - \sqrt{89}\right) < F_1^{s/a} < \frac{3}{8}\left(-13 + \sqrt{89}\right)$ if we want to simultaneously have $1 + F_1^{s/a}/3 < 0$ and $\delta\mathcal{F} \geq 0$. This implies that, in the presence of a gauge, the fundamental carriers of the gauge field Bose condense and subsequently lead to the breakdown of Galilean invariance [69]. Otherwise, the effective mass of the Landau quasiparticle $m^*/m = 1 + F_1^s/3$ could become negative, resulting in a Fermi surface instability and hence a contradiction [70]. The breakdown of Galilean invariance is of fundamental importance to our study, otherwise the addition of a spin-independent gauge field wouldn't affect the stability of the system. At a fundamental level, the introduction of either a natural or synthetic electromagnetic interaction through a gauge principle is well-known to be incompatible with Galilean field theories [71]. The argument from condensation is a direct consequence of a nonintegrable phase attached to the electron field operators (as previously noted in the context of Goldstone boson condensation [72]), and can be viewed as a manifestation of the Aharonov-Bohm effect [73]. As in $^3$He-$^4$He mixtures, a finite superfluid velocity defines a preferred frame and Galilean invariance of the fermionic system is subsequently lost. The fermionic system we consider can therefore be viewed as a non-Galilean-invariant Fermi liquid. Such a variation of the traditional Landau-Fermi liquid has already been noted to exist in the presence of a lattice geometry when the liquid phase exhibits low carrier concentrations [74].

A stronger argument for this condensation can be made by mapping the gauge bosons in our system to the Goldstone excitations in a macroscopic ring topology. In the metal-insulator transition of a topologically-trivial metal, condensation of the phonon mode in the dual geometry causes a distortion which breaks ring periodicity, producing a finite current in the simply connected space when we take the thermodynamic limit [75]. In this way, the condensation of the gauge field induces the formation of a metallic phase via symmetry breaking in the ordered state of the dual ring. The presence of finite-$\mathbf{A}$ dependence in the system's energy would then signify a metallic phase, while the absence of any gauge field-dependence would signify an insulating state [54, 75]. Our calculations of $\ell = 0$ and $\ell = 1$ Fermi surface stabilities with a finite gauge field can be seen as a manifestation of this phenomenon in a Landau-Fermi liquid.

Finally, we would like to point out a subtlety in the above calculations concerning the appearance of apparent dynamical *stabilities* brought about by a finite gauge. In the limit of small Fermi surface distortion, the stability condition determined by Eqns. (4)+(5) reduces to $1 + \frac{4}{27}F_1^{s/a} \geq 0$. When this bound is broken, the Fermi liquid ground state is stable only for some non-zero values of $\nu_{0\,s/a} \neq 0$. A calculation of the Drude weight $D_{s/a}$

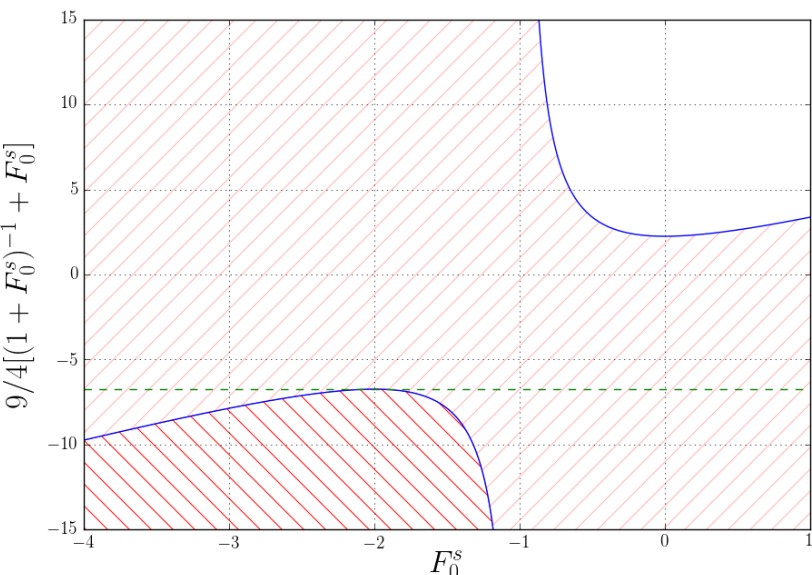

Figure 1: Visual representation of the Pomeranchuk instability for the symmetric gauge field (Eqn. (6a)) for $\nu_{0a} = 0$. In a stable Landau-Fermi liquid, $F_1^s$ is forbidden to enter the dark-red left-hatched region, while the light-red right-hatched region is characterized by a dynamical instability. For $\nu_{0s} = 0$, the value of $F_1^s$ is constrained to be above $-27/4$ (green dashed line).

yields a similar expression, with the Kohn relation [54, 55, 76] yielding

$$
\begin{aligned}
D_{s/a} &\equiv \left. \frac{\partial^2 E(\mathbf{A}_\sigma)}{\partial \mathbf{A}_\sigma^2} \right|_{\mathbf{A}_\sigma = 0} \\
&= \frac{9\pi n}{m^*} \left( 1 + \frac{4}{27} F_1^{s/a} \right)
\end{aligned}
\tag{7}
$$

where $E(\mathbf{A}_\sigma)$ is the energy density. While a positive Drude weight signals a stable metallic phase, a negative Drude weight for the symmetric gauge field implies an unstable maximum and an unusual paramagnetic response [77]. For the anti-symmetric gauge, the negative Drude weight implies a negative spin stiffness and hence an instability away from ferromagnetic ordering [78]. These results suggest that that systems with $F_1^{s/a} \leq -\frac{27}{4}$ are marked by instabilities not captured by the above analysis.

We interpret the dynamic fragility of the $\mathbf{A}_\sigma \neq 0$ Fermi liquid phase for $F_1^{s/a} \leq -\frac{27}{4}$ as a violation of Mermin's theorem on the sufficient conditions for the propagation of zero sound [79]. In a nutshell, Mermin's seminal work relies upon four ingredients: (1) the stability condition Eqn. (2); (2) the Landau kinetic equation; (3) the relation between the forward scattering amplitude $a_{pp'}^{s/a}$ and the Landau parameter:

$$
a_{pp'}^{s/a} = f_{pp'} + \sum_{p''} f_{pp''}^{s/a} \frac{\partial n_{p''}^0}{\partial \epsilon_{p''}} a_{p''p'}^{s/a}
\tag{8}
$$

and (4) the exclusion principle, which requires that $a_{pp}^s + a_{pp}^a = 0$. With these four conditions, it can be shown that the Landau kinetic equation has at least one eigenvalue

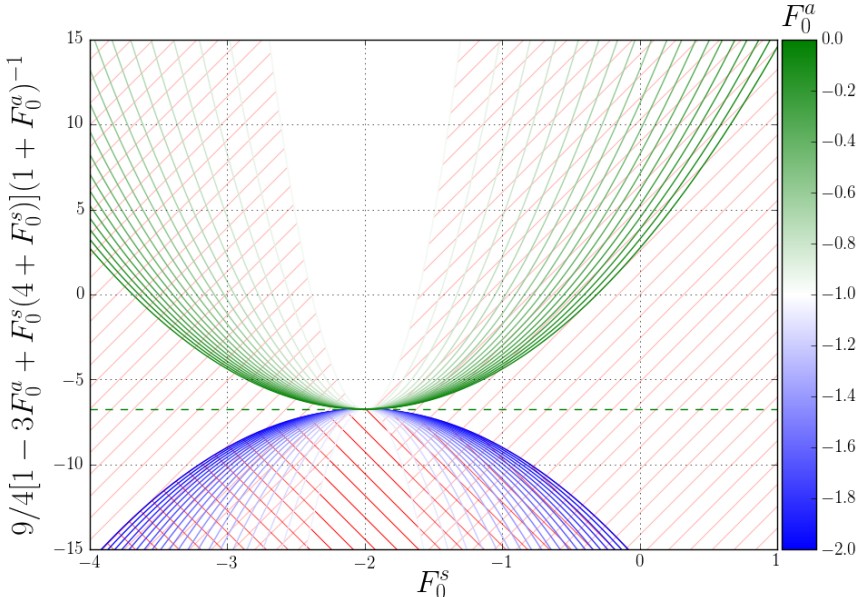

Figure 2: Visual representation of the Pomeranchuk instability for the anti-symmetric gauge field (Eqn. (6b)) for $\nu_{0s} = 0$. For $F_0^a > -1$, the Fermi liquid experiences a dynamical instability when $F_1^a$ is below the solid green line. For $F_0^a < -1$, phase separation occurs when $F_1^a$ is below the solid blue line. For $\nu_{0s} = 0$, the value of $F_1^s$ is constrained to be above $-27/4$ (green dashed line).

$\lambda$ of unity, and hence forward scattering is sufficient for the existence of regular or spin zero sound in a Fermi liquid [79]. In the presence of a gauge, the forward scattering amplitude and the exclusion principle are identical as in the $\mathbf{A}_\sigma = 0$ case. The stability for $\mathbf{A}_\sigma \neq 0$ has been discussed previously, and the previously homogenous kinetic equation is modified to include a term proportional to $\frac{\cos\theta}{\eta - \cos\theta}\left(\mathbf{A}_\sigma \cdot \frac{\mathbf{p}}{m}\right)$. Combining this with the forward scattering sum rule, the kinetic equation becomes

$$\nu = \left(\frac{\cos\theta}{\eta Y} + a^{s/a}\right)\nu + \cos\theta\left(\frac{(Y-1)\nu + X}{Y\eta}\right)\left\{\frac{1 - f^{s/a}}{1 + f^{s/a}}\right\} \tag{9}$$

where $\eta$ is the dimensionless phase velocity of zero sound, $Y \equiv a_{p,\,p-A}^{\sigma\sigma}\frac{\partial n_p^0}{\partial\epsilon_p}$, $X \approx \frac{A_\sigma p_F}{m}\cos\theta$, and the notation $\sum_{p'} B_{pp'}\frac{\partial n_{p'}^0}{\partial\epsilon_{p'}}\nu_{p'} \equiv -B\nu$ is taken for some observable $B = a_{pp'}^{s/a}$ or $f_{pp'}^{s/a}$. In the limit $A_\sigma << p_F$ and $\eta \to \infty$, Eqn. (9) reduces to $\nu = a^{s/a}\nu$ as in the original formulation. In the gauge-free theory $a_{pp'}^{s/a}$ is bounded to be below one from Eqn. (2). By calculating the terms in the kinetic equation, one can show [79] that there exists an $\eta'$ such that $\lambda_{\eta'} > 1$, implying that there exists some $\eta''$ between $\eta'$ and $\infty$ where $\lambda_{\eta''} = 1$. In the presence of a finite gauge, however, the modification of the Pomeranchuk condition results in the forward scattering amplitude taking values greater than one for $1 + F_0^{s/a} < 0$. This implies that forward scatting is not a sufficient condition for the propagation of zero sound if a weak gauge field is present. The onset of this instability can be seen explicitly by calculating the terms of interest in Eqn. (9):

$$\cos\theta\nu \equiv \eta^{-1}\int\frac{d\hat{n}}{4\pi}|\chi(\hat{n})|^2\cos\theta = \frac{1}{2} - \frac{\eta^{-1}\mathbb{A}^2}{2}\int_{\cos\theta_0}^1\frac{dx}{\eta - x} \tag{10a}$$

$$B\nu \equiv \int \frac{d\hat{n}}{4\pi} \int \frac{d\hat{n}'}{4\pi} \chi^*(\hat{n}) B \chi(\hat{n}')$$

$$= \frac{B\mathbb{A}^2}{4} \iint_{\cos\theta_0}^{1} \frac{dx}{\eta - x} \frac{dx'}{\eta - x'} \tag{10b}$$

where we have used the ansatz $\chi(\hat{n}) = \mathbb{A}/(\eta - \cos\theta)$, with $\mathbb{A}$ taken as a normalization constant and $0 < \theta < \theta_0$. As we approach $\eta = 1$ from above, the coefficient of $B$ diverges. Rearranging Eqn. (9), we therefore find a constraint on the value of $\nu$ assuming some finite gauge:

$$\nu < \frac{X}{\eta} \cos\theta \left( \frac{\cos\theta}{\eta} + a^{s/a} \right)^{-1} \tag{11}$$

The existence of zero sound for $\mathbf{A}_\sigma \neq 0$ then requires that (1) the right-hand side of Eqn. (9) yields a solution $< 1$ for some $\eta$ and (2) $\nu$ doesn't break the bound set by Eqn. (11). As the Fermi surface is only stable for larger, finite $\nu$ when $F_1^{s/a} < -27/4$, the calculation above suggests any metallic phase in this regime will be fragile to the disappearance of zero sound and subsequently a breakdown of conventional Landau-Fermi liquid theory in agreement with the Drude weight result.

In summary, we have investigated Fermi surface and zero sound instabilities in a Landau-Fermi liquid with weak, non-zero spin symmetric and spin anti-symmetric gauge fields $\mathbf{A}_\sigma$. Regardless of the magnitude of this gauge, we find that all Fermi liquids with $1 + F_0^{s/a} < 0$ are stable assuming that $1 + \frac{4}{27} F_1^{s/a} > 0$ and $\nu_{\ell, s/a}$ does not differ appreciably from zero. As a consequence, Landau-Fermi liquid theory is consistent with an itinerant form of ferromagnetism independent of phase separation as long as $\mathbf{A}_\sigma \neq 0$. Such a result complements the work done in [44, 46] on Fermi surface instabilities for partial waves $\ell > 0$; whereas these studies suggest an $\ell = 1$ Pomeranchuk instability is forbidden under certain conditions of the form factor where high- and low-energy contributions to the susceptibility are directly correlated, our work shows $\ell = 0$ instabilities are eliminated in the presence of weak gauge fields and where non-quasiparticle contributions are absent from the response. The robust metallic phase we find is a clear manifestation of gauge boson condensation previously introduced in the context of a metal-insulator transition [75]. Experimentally, our work provides a physical explanation for the violation of the Stoner model in ultracold gases of $^6$Li [26–28], as the degenerate mixture is confined to the lowest hyperfine states as the result of Zeeman splitting in a weak magnetic field and, hence, a spin-symmetric gauge potential. Our work similarly compliments existing studies of 2D spin-orbit coupled Fermi gases in a harmonic potential, where a Rashba-type gauge has been predicted to stabilize itinerant ferromagnetism via modification of the single-particle dispersion [80, 81].

Overall, the main message of this paper is a positive one: that a major ingredient for spintronic devices (i.e., a spin-dependent gauge field) serves to stabilize the itinerant ferromagnetism required for coherent spin manipulation. Our results strongly suggest that optically-trapped gases of neutral atoms such as $^6$Li [82] and $^{40}$K [83–85] are ideal candidates for spintronic hardware, as the resultant non-Abelian gauge fields [10,52,62,86–92] in dimensions $D \geq 2$ legitimize a description of spin transport in the well-established language of Landau-Fermi liquid theory while simultaneously permitting maximal control over the gauge field itself.

*Acknowledgements.*–One of the authors (J.T.H.) thanks Matthew Gochan for useful feedback. We would also like to acknowledge Shou-Cheng Zhang for his work on unconventional ferromagnetism, which served as one of the catalysts for this project. This research was partially supported by the John H. Rourke endowment fund at Boston College.

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

**Supplemental Material:**

**Explicit calculation of the Pomeranchuk instability condition at finite gauge**

In this section, we will briefly go over the calculation of Eqns. (4) and (5). We begin with Eqn. (4), where we explicitly write down the first term in the Landau free energy:

$$\sum_{p\sigma}(\epsilon_{p-A_\sigma,\sigma} - \mu)\delta n_{p-A_\sigma,\sigma}$$

$$\approx v_F \sum_{p\sigma}(p - p_F - A_\sigma)\Bigg\{ -\delta p_F\delta(p_F - p) - A_\sigma\delta(p_F - p) - \frac{1}{2}\delta p_F^2\frac{\partial}{\partial p}\delta(p_F - p) - A_\sigma\delta p_F\frac{\partial}{\partial p}\delta(p_F - p)$$

$$-\frac{1}{2}A_\sigma^2\frac{\partial}{\partial p}\delta(p_F - p)\Bigg\}$$

$$\approx -v_F \sum_{p\sigma}\Bigg\{ (p - p_F)\delta p_F\delta(p_F - p) + \frac{1}{2}\delta p_F^2(p - p_F)\frac{\partial}{\partial p}\delta(p_F - p)$$

$$+ A_\sigma\Bigg[ -\delta p_F\delta(p_F - p) + (p - p_F)\delta(p_F - p) - \frac{1}{2}\delta p_F^2\frac{\partial}{\partial p}\delta(p_F - p) + \delta p_F(p - p_F)\frac{\partial}{\partial p}\delta(p_F - p)\Bigg]$$

$$+ A_\sigma^2\Bigg[ -\delta(p_F - p) - \delta p_F\frac{\partial}{\partial p}\delta(p_F - p) + \frac{1}{2}(p - p_F)\frac{\partial}{\partial p}\delta(p_F - p)\Bigg]\Bigg\} \tag{12}$$

where we ignore terms cubic in $A_\sigma$ and higher. The first terms are just the gauge-independent terms:

(1)

$$-v_F \sum_{p\sigma}(p - p_F)\delta p_F\delta(p_F - p) \rightarrow -\frac{v_F}{(2\pi\hbar)^3}\sum_\sigma\int d\Omega\,\delta p_F\int dp\,p^2(p - p_F)\delta(p - p_F) = \boxed{0} \tag{13}$$

(2)

$$-\frac{v_F}{2}\sum_{p\sigma}\delta p_F^2(p - p_F)\frac{\partial}{\partial p}\delta(p_F - p) \rightarrow$$

$$-\frac{v_F}{2(2\pi\hbar)^3}\sum_\sigma\int d\Omega\delta p_F^2\int dp\,p^2(p - p_F)\frac{\partial}{\partial p}\delta(p_F - p)$$

$$= -\frac{v_F}{2(2\pi\hbar)^3}\sum_\sigma\int d\Omega\,\delta p_F^2\left\{ p^2(p - p_F)\delta(p_F - p)\Big|_0^\infty - \int dp\,(3p^2 - 2pp_F)\delta(p_F - p)\right\}$$

$$= \frac{v_F p_F^2}{2(2\pi\hbar)^3}\sum_\sigma\int d\Omega\,\delta p_F^2$$

$$= \frac{N(0)}{4}\sum_\sigma\sum_{\ell_1,\ell_2}\nu_{\ell_1\sigma}\nu_{\ell_2\sigma}\int_{-1}^1\frac{d(\cos\theta)}{2}P_{\ell_1}(\cos\theta)P_{\ell_2}(\cos\theta)$$

$$= \boxed{\frac{N(0)}{4}\sum_{\ell\sigma}\frac{\nu_{\ell\sigma}^2}{2\ell + 1}} \tag{14}$$

These are the terms that remain when we take $A_\sigma = 0$. Now we will consider those that are linear in $A_\sigma$:

$$-v_F \sum_{p\sigma} A_\sigma \left[ -\delta p_F \delta(p_F - p) + (p - p_F)\delta(p_F - p) - \frac{1}{2}\delta p_F^2 \frac{\partial}{\partial p}\delta(p_F - p) + \delta p_F(p - p_F)\frac{\partial}{\partial p}\delta(p_F - p) \right]$$

(15)

(1)

$$v_F \sum_{p\sigma} \delta p_F \delta(p_F - p) \rightarrow \frac{v_F}{(2\pi\hbar)^3} \sum_\sigma \int d\Omega \delta p_F \int dp\, p^2 \delta(p_F - p) = \boxed{\frac{N(0)v_F}{2} \sum_\sigma \frac{\nu_{\ell\sigma}}{2\ell+1}\delta_{\ell 0}}$$

(16)

(2)

$$-v_F \sum_{p\sigma} (p - p_F)\delta(p_F - p) \rightarrow -\frac{v_F}{(2\pi\hbar)^3} \sum_\sigma \int d\Omega \int dp\, p^2 (p - p_F)\delta(p_F - p) = \boxed{0} \quad (17)$$

(3)

$$\frac{v_F}{2} \sum_{p\sigma} \delta p_F^2 \frac{\partial}{\partial p}\delta(p_F - p) \rightarrow \frac{v_F}{2(2\pi\hbar)^3} \sum_\sigma \int d\Omega\, \delta p_F^2 \int dp\, p^2 \frac{\partial}{\partial p}\delta(p_F - p) = \boxed{-\frac{N(0)}{2p_F} \sum_{\ell\sigma} \frac{\nu_{\ell\sigma}^2}{2\ell+1}}$$

(18)

(4)

$$-v_F \sum_{p\sigma} \delta p_F(p - p_F)\frac{\partial}{\partial p}\delta(p_F - p) = \boxed{\frac{N(0)v_F}{2} \sum_{\ell\sigma} \frac{\nu_{\ell\sigma}}{2\ell+1}\delta_{\ell 0}} \quad (19)$$

The term that is linear in $A_\sigma$ then becomes

$$\frac{N(0)}{4} \sum_{\ell\sigma} \frac{A_\sigma}{2\ell+1} \left\{ 4v_F \nu_{\ell\sigma}\delta_{\ell 0} - 2\nu_{\ell\sigma}^2/p_F \right\}$$

$$= \frac{N(0)}{4} \sum_\ell \left[ \frac{A_\uparrow}{2\ell+1}\left(4v_F\nu_{\ell\uparrow} - 2\nu_{\ell\uparrow}^2/p_F\right) + \frac{A_\downarrow}{2\ell+1}\left(4v_F\nu_{\ell\uparrow} - 2\nu_{\ell\downarrow}^2/p_F\right) \right]$$

$$= \begin{cases} \boxed{\dfrac{N(0)}{4}\sum_\ell \left[\dfrac{A}{2\ell+1}\left(4v_F\nu_{\ell s} - \dfrac{1}{p_F}\left(\nu_{\ell s}^2 + \nu_{\ell a}^2\right)\right)\right]}, & A_\uparrow = A_\downarrow \\[4mm] \boxed{\dfrac{N(0)}{4}\sum_\ell \left[\dfrac{A}{2\ell+1}\left(4v_F\nu_{\ell a} - \dfrac{2}{p_F}\nu_{\ell s}\nu_{\ell a}\right)\right]}, & A_\uparrow = -A_\downarrow \end{cases}$$

(20)

Where we have used the fact that

$$\nu_{\ell\uparrow}^2 + \nu_{\ell\downarrow}^2 = \frac{1}{2}\left(\nu_{\ell s}^2 + \nu_{\ell a}^2\right) \quad (21)$$

$$\nu_{\ell\uparrow}^2 - \nu_{\ell\downarrow}^2 = (\nu_{\ell\uparrow} + \nu_\downarrow)(\nu_{\ell\uparrow} - \nu_{\ell\downarrow}) = \nu_{\ell s}\nu_{\ell a} \tag{22}$$

We will now deal with quadratic terms:

$$-v_F \sum_{p\sigma} A_\sigma^2 \left[ -\delta(p_F - p) - \delta p_F \frac{\partial}{\partial p}\delta(p_F - p) + \frac{1}{2}(p - p_F)\frac{\partial}{\partial p}\delta(p_F - p) \right] \tag{23}$$

(1)

$$v_F \sum_{p\sigma} \delta(p_F - p) \rightarrow \frac{v_F}{(2\pi\hbar)^3} \sum_{p\sigma} \int d\Omega \int dp\, p^2 \delta(p_F - p) = \boxed{\sum_\sigma \frac{N(0)v_F^2}{2}} \tag{24}$$

(2)

$$v_F \sum_{p\sigma} \delta p_F \frac{\partial}{\partial p}\delta(p_F - p) \rightarrow \frac{v_F}{(2\pi\hbar)^3} \sum_{p\sigma} \int d\Omega \delta p_F \int dp\, p^2 \frac{\partial}{\partial p}\delta(p_F - p) = \boxed{-\frac{N(0)v_F}{p_F} \sum_{\ell\sigma} \frac{\nu_{\ell\sigma}}{2\ell+1}\delta_{\ell 0}} \tag{25}$$

(3)

$$-\frac{v_F}{2} \sum_{p\sigma} (p - p_F)\frac{\partial}{\partial p}\delta(p_F - p) \rightarrow -\frac{v_F}{2(2\pi\hbar)^3} \sum_\sigma \int d\Omega \int dp\, p^2 (p - p_F)\frac{\partial}{\partial p}\delta(p_F - p) = \boxed{\sum_\sigma \frac{N(0)v_F^2}{4}} \tag{26}$$

The quadratic terms are therefore given below:

$$\frac{N(0)}{4} \sum_\sigma A_\sigma^2 \left\{ 3v_F^2 - \sum_\ell \frac{4v_F \nu_{\ell\sigma}/p_F}{2\ell+1}\delta_{\ell 0} \right\} \tag{27}$$

Note that the above can be simplified as follows:

$$\frac{N(0)}{4} \sum_\sigma A_\sigma^2 \left\{ 3v_F^2 - \sum_\ell \frac{4v_F \nu_{\ell\sigma}/p_F}{2\ell+1}\delta_{\ell 0} \right\} = \frac{N(0)}{4} A^2 \left[ 6v_F^2 - \sum_\ell \frac{4v_F}{p_F(2\ell+1)}\nu_{\ell s}\delta_{\ell 0} \right] \tag{28}$$

Because we only care about terms up to quadratic in $A$, the above becomes

$$\sum_{p\sigma}(\epsilon_{p-A_\sigma,\sigma} - \mu)\delta n_{p-A_\sigma,\sigma}$$

$$\rightarrow \begin{cases} \frac{N(0)}{4}\left[\sum_\ell \frac{1}{2\ell+1}\left\{\frac{1}{2}(\nu_{\ell s}^2 + \nu_{\ell a}^2) + A\left\{4v_F\nu_{\ell s} - \frac{1}{p_F}\left(\nu_{\ell s}^2 + \nu_{\ell a}^2\right)\right\} - \frac{4A^2}{m^*}\nu_{\ell s}\delta_{\ell 0}\right\} + 6v_F^2 A^2\right], & A_\uparrow = A_\downarrow \\[3ex] \frac{N(0)}{4}\left[\sum_\ell \frac{1}{2\ell+1}\left\{\frac{1}{2}(\nu_{\ell s}^2 + \nu_{\ell a}^2) + 2A\nu_{\ell a}\left\{2v_F - \frac{\nu_{\ell s}}{p_F}\right\} - \frac{4A^2}{m^*}\nu_{\ell s}\delta_{\ell 0}\right\} + 6v_F^2 A^2\right], & A_\uparrow = -A_\downarrow \end{cases}$$

$$= \begin{cases} \frac{N(0)}{4}\left[\sum_\ell \frac{1}{2\ell+1}\left\{\frac{1}{2}(\nu_{\ell s}^2 + \nu_{\ell a}^2) + \frac{A}{p_F}\left\{8\epsilon_F\nu_{\ell s} - \left(\nu_{\ell s}^2 + \nu_{\ell a}^2\right)\right\} - 8\frac{A^2\epsilon_F}{p_F^2}\nu_{\ell s}\delta_{\ell 0}\right\} + \frac{24\epsilon_F^2 A^2}{p_F^2}\right], & A_\uparrow = A_\downarrow \\[3ex] \frac{N(0)}{4}\left[\sum_\ell \frac{1}{2\ell+1}\left\{\frac{1}{2}(\nu_{\ell s}^2 + \nu_{\ell a}^2) + 2\frac{A}{p_F}\nu_{\ell a}\left\{4\epsilon_F - \nu_{\ell s}\right\} - 8\frac{A^2\epsilon_F}{p_F^2}\nu_{\ell s}\delta_{\ell 0}\right\} + \frac{24\epsilon_F^2 A^2}{p_F^2}\right], & A_\uparrow = -A_\downarrow \end{cases}$$

$$\approx \boxed{\begin{cases} \frac{N(0)}{8}\left[\sum_\ell \frac{1}{2\ell+1}\left\{(\nu_{\ell s}^2 + \nu_{\ell a}^2) + \frac{16A}{p_F}\epsilon_F\nu_{\ell s}\right\} + \frac{48A^2}{p_F^2}\epsilon_F^2\right], & A_\uparrow = A_\downarrow \\[3ex] \frac{N(0)}{8}\left[\sum_\ell \frac{1}{2\ell+1}\left\{(\nu_{\ell s}^2 + \nu_{\ell a}^2) + \frac{16A}{p_F}\epsilon_F\nu_{\ell a}\right\} + \frac{48A^2}{p_F^2}\epsilon_F^2\right], & A_\uparrow = -A_\downarrow \end{cases}}$$

where we have used the fact that $\epsilon_F >> \nu_s$ and $\epsilon_F >> \nu_a$. This completes the derivation of Eqn. (4).

We will now consider the gauge-dependence of the quadratic term in the Landau-Fermi free energy; i.e., the derivation of Eqn. (5):

$$\sum_{pp'\sigma\sigma'} f_{pp'\sigma\sigma'} \delta n_{p-A_\sigma,\sigma} \delta n_{p'-A_{\sigma'},\sigma'}$$

$$= \sum_{pp'\sigma\sigma'} f_{pp'\sigma\sigma'}$$

$$\times \left\{ -\delta p_F \delta(p_F - p) - A_\sigma \delta(p_F - p) - \frac{1}{2}\delta p_F^2 \frac{\partial}{\partial p}\delta(p_F - p) - A_\sigma \delta p_F \frac{\partial}{\partial p}\delta(p_F - p) - \frac{1}{2}A_\sigma^2 \frac{\partial}{\partial p}\delta(p_F - p) \right\}$$

$$\times \left\{ -\delta p_F' \delta(p_F - p') - A_{\sigma'} \delta(p_F - p') - \frac{1}{2}\delta p_F'^2 \frac{\partial}{\partial p'}\delta(p_F - p') - A_{\sigma'} \delta p_F' \frac{\partial}{\partial p'}\delta(p_F - p') - \frac{1}{2}A_{\sigma'}^2 \frac{\partial}{\partial p'}\delta(p_F - p') \right\}$$

$$\approx \sum_{pp'\sigma\sigma'} f_{pp'\sigma\sigma'} \left\{ \delta p_F \delta p_F' \delta(p_F - p)\delta(p_F - p') \right.$$

$$+ A_\sigma \left[ \delta p_F' \delta(p_F - p')\delta(p_F - p) + \delta p_F \delta p_F' \delta(p_F - p')\frac{\partial}{\partial p}\delta(p_F - p) + \frac{1}{2}\delta p_F'^2 \delta(p_F - p)\frac{\partial}{\partial p'}\delta(p_F - p') \right]$$

$$+ A_{\sigma'} \left[ \delta p_F \delta(p_F - p)\delta(p_F - p') + \delta p_F' \delta p_F \delta(p_F - p)\frac{\partial}{\partial p'}\delta(p_F - p') + \frac{1}{2}\delta p_F^2 \delta(p_F - p')\frac{\partial}{\partial p}\delta(p_F - p) \right]$$

$$+ A_\sigma^2 \left[ \frac{1}{2}\delta p_F' \delta(p_F - p')\frac{\partial}{\partial p}\delta(p_F - p) + \frac{1}{4}\delta p_F'^2 \frac{\partial}{\partial p'}\delta(p_F - p')\frac{\partial}{\partial p}\delta(p_F - p) \right]$$

$$+ A_{\sigma'}^2 \left[ \frac{1}{2}\delta p_F \delta(p_F - p)\frac{\partial}{\partial p'}\delta(p_F - p') + \frac{1}{4}\delta p_F^2 \frac{\partial}{\partial p}\delta(p_F - p)\frac{\partial}{\partial p'}\delta(p_F - p') \right]$$

$$+ A_\sigma \cdot A_{\sigma'} \left[ \delta(p_F - p)\delta(p_F - p') + \delta p_F' \delta(p_F - p)\frac{\partial}{\partial p'}\delta(p_F - p') \right.$$

$$\left. + \delta p_F \delta(p_F - p')\frac{\partial}{\partial p}\delta(p_F - p) + \delta p_F \delta p_F' \frac{\partial}{\partial p}\delta(p_F - p)\frac{\partial}{\partial p'}\delta(p_F - p') \right] \tag{29}$$

where we only keep terms up to quadratic dependence in $A_\sigma$. To begin, we look at the gauge-independent term:

(1)

$$\sum_{pp'\sigma\sigma'} f_{pp'\sigma\sigma'} \delta p_F \delta p'_F \delta(p_F - p)\delta(p_F - p')$$

$$\rightarrow \sum_{\sigma\sigma'} \frac{1}{(2\pi\hbar)^6} \int d\Omega \int d\Omega' \int dp\, p^2 \int dp'\, p'^2 f_{pp'\sigma\sigma'} \delta(p_F - p)\delta(p_F - p')\delta p_F \delta p'_F$$

$$= \frac{p_F^4}{(2\pi\hbar)^6} \sum_{\sigma\sigma'} \int d\Omega \delta p_F \int d\Omega' \delta p'_F f_{pp'\sigma\sigma'}$$

$$= \frac{16\pi^2 p_F^4}{(2\pi\hbar)^6 v_F^2} \sum_{\ell\ell'\ell''\sigma\sigma'} \nu_{\ell\sigma}\nu_{\ell'\sigma'} \int_{-1}^{1} \frac{d(\cos\theta)}{2} \int_{-1}^{1} \frac{d(\cos\theta')}{2} f_{\ell''\sigma'\sigma''} P_\ell(\cos\theta) P_{\ell'}(\cos\theta') P_{\ell''}(\cos\theta') P_{\ell''}(\cos\theta)$$

$$= \frac{N(0)}{4} \sum_{\ell\sigma\sigma'} \nu_{\ell\sigma}\nu_{\ell\sigma'} \frac{1}{2\ell+1} \frac{F_{\ell\sigma\sigma'}}{2\ell+1}$$

$$= \frac{N(0)}{4} \sum_{\ell} \frac{1}{2\ell+1} \frac{1}{2\ell+1} \left\{ \nu_{\ell\uparrow}^2 F_{\ell\uparrow\uparrow} + 2\nu_{\ell\uparrow}\nu_{\ell\downarrow} F_{\ell\uparrow\downarrow} + \nu_{\ell\downarrow}^2 F_{\ell\downarrow\downarrow} \right\}$$

$$= \boxed{\frac{N(0)}{4} \sum_{\ell} \frac{1}{2\ell+1} \frac{1}{2\ell+1} \left\{ F_\ell^s \nu_{\ell s}^2 + F_\ell^a \nu_{\ell a}^2 \right\}} \tag{30}$$

We will now calculate the terms linear in $A_\sigma$:

$$\sum_{pp'\sigma\sigma'} f_{pp'\sigma\sigma'}$$

$$\times \left\{ A_\sigma \left[ \delta p'_F \delta(p_F - p')\delta(p_F - p) + \delta p_F \delta p'_F \delta(p_F - p') \frac{\partial}{\partial p} \delta(p_F - p) + \frac{1}{2} \delta p_F'^2 \delta(p_F - p) \frac{\partial}{\partial p'} \delta(p_F - p') \right] \right.$$

$$\left. + A_{\sigma'} \left[ \delta p_F \delta(p_F - p)\delta(p_F - p') + \delta p'_F \delta p_F \delta(p_F - p) \frac{\partial}{\partial p'} \delta(p_F - p') + \frac{1}{2} \delta p_F^2 \delta(p_F - p') \frac{\partial}{\partial p} \delta(p_F - p) \right] \right\}$$

$$= 2 \sum_{pp'\sigma\sigma'} f_{pp'\sigma\sigma'}$$

$$\times \left\{ A_\sigma \left[ \delta p'_F \delta(p_F - p')\delta(p_F - p) + \delta p_F \delta p'_F \delta(p_F - p') \frac{\partial}{\partial p} \delta(p_F - p) + \frac{1}{2} \delta p_F'^2 \delta(p_F - p) \frac{\partial}{\partial p'} \delta(p_F - p') \right] \right\} \tag{31}$$

(1)

$$\sum_{pp'\sigma\sigma'} f_{pp'\sigma\sigma'} \delta p'_F \delta(p_F - p')\delta(p_F - p) = \boxed{\frac{N(0)v_F}{4} \sum_{\ell'\sigma\sigma'} \nu_{\ell'\sigma'} \frac{1}{2\ell'+1} \frac{F_{\ell'\sigma\sigma'}}{2\ell'+1} \delta_{\ell'0}} \tag{32}$$

(2)

$$\sum_{pp'\sigma\sigma'} f_{pp'\sigma\sigma'} \delta p_F \delta p'_F \delta(p_F - p') \frac{\partial}{\partial p} \delta(p_F - p) = \boxed{-\frac{N(0)}{2p_F} \sum_{\ell\sigma\sigma'} \nu_{\ell\sigma}\nu_{\ell\sigma'} \frac{1}{2\ell+1} \frac{F_{\ell\sigma\sigma'}}{2\ell+1}} \tag{33}$$

(3)

$$\frac{1}{2}\sum_{pp'\sigma\sigma'}f_{pp'\sigma\sigma'}\delta p_F'^2\delta(p_F-p)\frac{\partial}{\partial p'}\delta(p_F-p')$$

$$\rightarrow\frac{1}{2(2\pi\hbar)^6}\sum_{\sigma\sigma'}\int d\Omega\int d\Omega'\int dp\,p^2\int dp'\,p'^2 f_{pp'\sigma\sigma'}\delta p_F'^2\delta(p_F-p)\frac{\partial}{\partial p'}\delta(p_F-p')$$

$$=-\frac{16\pi^2 p_F^3}{(2\pi\hbar)^6 v_F^2}\sum_{\ell\ell'\ell''\sigma\sigma'}\nu_{\ell\sigma'}\nu_{\ell'\sigma'}f_{\ell''\sigma\sigma'}\int_{-1}^{1}\frac{d(\cos\theta)}{2}P_{\ell''}(\cos\theta)\int_{-1}^{1}\frac{d(\cos\theta')}{2}P_{\ell}(\cos\theta')P_{\ell'}(\cos\theta')P_{\ell''}(\cos\theta')$$

$$=-\frac{p_F^3}{4\pi^2\hbar^6 v_F^2}\sum_{\ell\ell'\ell''\sigma\sigma'}\nu_{\ell\sigma'}\nu_{\ell'\sigma'}f_{\ell''\sigma\sigma'}\frac{1}{2\ell''+1}\delta_{\ell''0}\begin{pmatrix}\ell & \ell' & \ell''\\0 & 0 & 0\end{pmatrix}^2$$

$$=\boxed{-\frac{N(0)}{4p_F}\sum_{\ell\sigma\sigma'}\nu_{\ell\sigma}^2\frac{F_{0\sigma\sigma'}}{2\ell+1}}\tag{34}$$

Where we have used the fact that

$$\begin{pmatrix}\ell & \ell' & 0\\0 & 0 & 0\end{pmatrix}=(-1)^{\frac{\ell+\ell'}{2}}\sqrt{\frac{(\ell'-\ell)!(\ell-\ell')!}{\ell+\ell'+1}}\frac{1}{\left(\frac{\ell'-\ell}{2}\right)!\left(\frac{\ell-\ell'}{2}\right)!}=-\frac{1}{\sqrt{2\ell+1}}\delta_{\ell\ell'}\tag{35}$$

We can now write the linear term in $A_\sigma$:

$$2\frac{N(0)}{4}\sum_{\ell\sigma\sigma'}A_\sigma\left[\nu_{\ell\sigma}\frac{v_F}{2\ell+1}\frac{F_{\ell\sigma\sigma'}}{2\ell+1}\delta_{\ell 0}-\frac{2}{p_F}\nu_{\ell\sigma}\nu_{\ell\sigma'}\frac{1}{2\ell+1}\frac{F_{\ell\sigma\sigma'}}{2\ell+1}-\frac{1}{p_F}\nu_{\ell\sigma}^2\frac{F_{0\sigma\sigma'}}{2\ell+1}\right]$$

$$=\frac{N(0)}{2}\sum_{\ell\sigma\sigma'}A_\sigma\frac{1}{2\ell+1}\left[\nu_{\ell\sigma}v_F\frac{F_{\ell\sigma\sigma'}}{2\ell+1}\delta_{\ell 0}-\frac{1}{p_F}\left\{2\nu_{\ell\sigma}\nu_{\ell\sigma'}\frac{F_{\ell\sigma\sigma'}}{2\ell+1}+\nu_{\ell\sigma}^2 F_{0\sigma\sigma'}\right\}\right]$$

$$=\frac{N(0)}{4}\sum_{\ell}\frac{1}{2\ell+1}\frac{1}{2\ell+1}\left[\sum_{\sigma\sigma'}2A_\sigma\nu_{\ell\sigma}v_F F_{\ell\sigma\sigma'}\delta_{\ell 0}-\sum_{\sigma\sigma'}A_\sigma\frac{2}{p_F}\left\{2\nu_{\ell\sigma}\nu_{\ell\sigma'}F_{\ell\sigma\sigma'}+\nu_{\ell\sigma}^2 F_{0\sigma\sigma'}(2\ell+1)\right\}\right]$$

$$\tag{36}$$

Simplifying the above cases piece by piece, we find the following:

(1)

$$\sum_{\sigma\sigma'}A_\sigma 2\nu_{\ell\sigma}v_F F_{\ell\sigma\sigma'}\delta_{\ell 0}=2A_\uparrow\nu_{\ell\uparrow}v_F\delta_{\ell 0}(F_{\ell\uparrow\uparrow}+F_{\ell\uparrow\downarrow})+2A_\downarrow\nu_{\ell\downarrow}v_F\delta_{\ell 0}(F_{\ell\downarrow\uparrow}+F_{\ell\downarrow\downarrow})$$

$$=\begin{cases}4v_F A F_\ell^s\nu_{\ell s}\delta_{\ell 0}, & A_\uparrow=A_\downarrow\\[2mm]4v_F A F_\ell^s\nu_{\ell a}\delta_{\ell 0}, & A_\uparrow=-A_\downarrow\end{cases}$$

(2)

$$\sum_{\sigma\sigma'}A_\sigma\frac{2}{p_F}\left\{2\nu_{\ell\sigma}\nu_{\ell\sigma'}F_{\ell\sigma\sigma'}+\nu_{\ell\sigma}^2 F_{0\sigma\sigma'}(2\ell+1)\right\}$$

$$=\begin{cases}\frac{4A}{p_F}\left\{\nu_{\ell s}^2\left[F_\ell^s+\frac{1}{2}(2\ell+1)F_0^s\right]+\nu_{\ell a}^2\left[F_\ell^a+\frac{1}{2}(2\ell+1)F_0^s\right]\right\}, & A_\uparrow=A_\downarrow\\[3mm]\frac{4A}{p_F}\nu_{\ell s}\nu_{\ell a}\left\{F_\ell^s+F_\ell^a+(2\ell+1)F_0^s\right\}, & A_\uparrow=-A_\downarrow\end{cases}$$

The term linear in $A_\sigma$ can then be written as the following:

$$\frac{N(0)}{4} \sum_\ell \frac{1}{2\ell+1} \frac{1}{2\ell+1} \left[ \sum_{\sigma\sigma'} 2A_\sigma \nu_{\ell\sigma} v_F F_{\ell\sigma\sigma'} \delta_{\ell 0} - \sum_{\sigma\sigma'} A_\sigma \frac{2}{p_F} \left\{ 2\nu_{\ell\sigma}\nu_{\ell\sigma'} F_{\ell\sigma\sigma'} + \nu_{\ell\sigma}^2 F_{0\sigma\sigma'}(2\ell+1) \right\} \right]$$

$$= \frac{N(0)}{4} \sum_\ell \frac{1}{2\ell+1} \frac{1}{2\ell+1}$$

$$\times 4A \begin{cases} v_F F_\ell^s \nu_{\ell s} \delta_{\ell 0} - \frac{1}{p_F} \left\{ \nu_{\ell s}^2 \left[ F_\ell^s + \frac{1}{2}(2\ell+1)F_0^s \right] + \nu_{\ell a}^2 \left[ F_\ell^a + \frac{1}{2}(2\ell+1)F_0^s \right] \right\}, & A_\uparrow = A_\downarrow \\[2mm] v_F F_\ell^s \nu_{\ell a} \delta_{\ell 0} - \frac{1}{p_F} \nu_{\ell s} \nu_{\ell a} \left\{ F_\ell^s + F_\ell^a + (2\ell+1)F_0^s \right\}, & A_\uparrow = -A_\downarrow \end{cases}$$

$$(37)$$

We will now tackle the terms that go as $A_\sigma^2$:

$$\sum_{pp'\sigma\sigma'} f_{pp'\sigma\sigma'} \left\{ A_\sigma^2 \left[ \frac{1}{2} \delta p'_F \delta(p_F - p') \frac{\partial}{\partial p} \delta(p_F - p) + \frac{1}{4} \delta p_F'^2 \frac{\partial}{\partial p'} \delta(p_F - p') \frac{\partial}{\partial p} \delta(p_F - p) \right] \right.$$

$$\left. + A_{\sigma'}^2 \left[ \frac{1}{2} \delta p_F \delta(p_F - p) \frac{\partial}{\partial p'} \delta(p_F - p') + \frac{1}{4} \delta p_F^2 \frac{\partial}{\partial p} \delta(p_F - p) \frac{\partial}{\partial p'} \delta(p_F - p') \right] \right\}$$

$$= \sum_{pp'\sigma\sigma'} f_{pp'\sigma\sigma'} \left\{ A_\sigma^2 \left[ \delta p'_F \delta(p_F - p') \frac{\partial}{\partial p} \delta(p_F - p) + \frac{1}{2} \delta p_F'^2 \frac{\partial}{\partial p'} \delta(p_F - p') \frac{\partial}{\partial p} \delta(p_F - p) \right] \right\}$$

$$(38)$$

Each term is calculated one-by-one:

(1)

$$\sum_{pp'\sigma\sigma'} f_{pp'\sigma\sigma'} \delta p'_F \delta(p_F - p') \frac{\partial}{\partial p} \delta(p_F - p) = \boxed{ -\frac{N(0)v_F}{2p_F} \sum_{\ell\ell'\sigma\sigma'} \nu_{\ell\sigma} \frac{1}{2\ell+1} \frac{1}{2\ell+1} \frac{F_{\ell\sigma\sigma'}}{2\ell+1} \delta_{\ell 0} } \quad (39)$$

(2)

$$\frac{1}{2} \sum_{pp'\sigma\sigma'} f_{pp'\sigma\sigma'} \delta p_F'^2 \frac{\partial}{\partial p'} \delta(p_F - p') \frac{\partial}{\partial p} \delta(p_F - p)$$

$$\rightarrow \frac{1}{2(2\pi\hbar)^6} \int d\Omega \int d\Omega' \int dp\, p^2 \int dp'\, p'^2\, f_{pp'\sigma\sigma'} \delta p_F'^2 \frac{\partial}{\partial p'} \delta(p_F - p') \frac{\partial}{\partial p} \delta(p_F - p)$$

$$= \frac{4p_F^2}{2(2\pi\hbar)^6} \int d\Omega \int d\Omega' \delta p_F'^2 f_{pp'\sigma\sigma'}$$

$$= \frac{64\pi^2 p_F^2}{2(2\pi\hbar)^6 v_F^2} \sum_{\ell\ell'\ell''\sigma\sigma'} \nu_{\ell'\sigma'} \nu_{\ell'\sigma'} f_{\ell''\sigma\sigma'} \int_{-1}^1 \frac{d(\cos\theta)}{2} \int_{-1}^1 \frac{d(\cos\theta')}{2} P_\ell(\cos\theta') P_{\ell'}(\cos\theta') P_{\ell''}(\cos\theta) P_{\ell''}(\cos\theta')$$

$$= \frac{p_F^2}{2\pi^4 \hbar^6 v_F^2} \sum_{\ell\ell'\ell''\sigma\sigma'} \nu_{\ell'\sigma'} \nu_{\ell'\sigma'} f_{\ell''\sigma\sigma'} \frac{1}{2\ell''+1} \delta_{\ell''0} \begin{pmatrix} \ell & \ell' & \ell'' \\ 0 & 0 & 0 \end{pmatrix}^2$$

$$= \boxed{ \frac{N(0)}{2p_F^2} \sum_{\ell\sigma\sigma'} \nu_{\ell\sigma}^2 \frac{F_{0\sigma\sigma'}}{2\ell+1} } \quad (40)$$

Thus, the above resulst in the following term in $A_\sigma^2$:

$$2 \sum_{\ell\sigma\sigma'} A_\sigma^2 \left\{ -\frac{N(0)}{2p_F} \nu_{\ell\sigma} \frac{1}{2\ell+1} \frac{F_{\ell\sigma\sigma'}}{2\ell+1} \delta_{\ell 0} + \frac{N(0)}{2p_F^2} \nu_{\ell\sigma}^2 \frac{F_{0\sigma\sigma'}}{2\ell+1} \right\}$$

$$=\frac{N(0)}{4} \sum_\ell \frac{1}{2\ell+1} \frac{1}{2\ell+1} 4 \sum_{\sigma\sigma'} A_\sigma^2 \left\{ -\frac{\nu_{\ell\sigma}}{m^*} F_{\ell\sigma\sigma'} \delta_{\ell 0} + \frac{\nu_{\ell\sigma}^2}{p_F^2} F_{0\sigma\sigma'}(2\ell+1) \right\} \qquad (41)$$

We look at each term separately:

$$-\sum_{\sigma\sigma'} A_\sigma^2 \frac{\nu_{\ell\sigma}}{m^*} F_{\ell\sigma\sigma'} \delta_{\ell 0} = -\frac{\delta_{\ell 0}}{m^*} \left\{ A_\uparrow^2 \nu_\uparrow F_{\ell\uparrow\uparrow} + A_\uparrow^2 \nu_\uparrow F_{\ell\uparrow\downarrow} + A_\downarrow^2 \nu_\downarrow F_{\ell\downarrow\uparrow} + A_\downarrow^2 \nu_\downarrow F_{\ell\downarrow\downarrow} \right\}$$

$$= -\frac{2}{m^*} A^2 \nu_{\ell s} F_\ell^s \delta_{\ell 0} \qquad (42)$$

$$\frac{2\ell+1}{p_F^2} \sum_{\sigma\sigma'} A_\sigma^2 \nu_{\ell\sigma}^2 F_{0\sigma\sigma'} = \frac{2\ell+1}{p_F^2} \left\{ A_\uparrow^2 \nu_{\ell\uparrow}^2 (F_{0\uparrow\uparrow} + F_{0\uparrow\downarrow}) + A_\downarrow^2 \nu_{\ell\downarrow}^2 (F_{0\downarrow\uparrow} + F_{0\downarrow\downarrow}) \right\}$$

$$= (2\ell+1)\frac{A^2}{p_F^2} F_0^s (\nu_{\ell s}^2 + \nu_{\ell a}^2) \qquad (43)$$

Hence, we find that the term that goes as $A_\sigma^2$ goes as

$$\frac{N(0)}{4} \sum_\ell \frac{1}{2\ell+1} \frac{1}{2\ell+1} 4 \sum_{\sigma\sigma'} A_\sigma^2 \left\{ -\frac{\nu_{\ell\sigma}}{m^*} F_{\ell\sigma\sigma'} \delta_{\ell 0} + \frac{\nu_{\ell\sigma}^2}{p_F^2} F_{0\sigma\sigma'}(2\ell+1) \right\}$$

$$=\boxed{\frac{N(0)}{4} \sum_\ell \frac{1}{2\ell+1} \frac{1}{2\ell+1} \left[ 4A^2 \left\{ -\frac{2}{m^*} \nu_{\ell s} F_\ell^s \delta_{\ell 0} + (2\ell+1)\frac{1}{p_F^2} F_0^s (\nu_{\ell s}^2 + \nu_{\ell a}^2) \right\} \right]} \qquad (44)$$

We will now deal with the terms that go as $A_\sigma \cdot A_{\sigma'}$:

$$\sum_{pp'\sigma\sigma'} f_{pp'\sigma\sigma'} A_\sigma \cdot A_{\sigma'} \left[ \delta(p_F - p)\delta(p_F - p') + \delta p_F' \delta(p_F - p)\frac{\partial}{\partial p'}\delta(p_F - p') \right.$$

$$\left. + \delta p_F \delta(p_F - p')\frac{\partial}{\partial p}\delta(p_F - p) + \delta p_F \delta p_F' \frac{\partial}{\partial p}\delta(p_F - p)\frac{\partial}{\partial p'}\delta(p_F - p') \right]$$

$$= \sum_{pp'\sigma\sigma'} f_{pp'\sigma\sigma'} A_\sigma \cdot A_{\sigma'}$$

$$\times \left[ \delta(p_F - p)\delta(p_F - p') + 2\delta p_F \delta(p_F - p')\frac{\partial}{\partial p}\delta(p_F - p) + \delta p_F \delta p_F' \frac{\partial}{\partial p}\delta(p_F - p)\frac{\partial}{\partial p'}\delta(p_F - p') \right]$$

$$(45)$$

We go through each term separately, being careful to include a $\cos\chi$ term due to the

dot product: (1)

$$\sum_{pp'\sigma\sigma'} f_{pp'\sigma\sigma'} \cos\theta \cos\theta' \delta(p_F - p)\delta(p_F - p')$$

$$= \frac{p_F^4}{4\pi^4\hbar^6} \sum_{\ell\sigma\sigma'} f_{\ell\sigma\sigma'} \int_{-1}^{1} \frac{d(\cos\theta)}{2} \cos\theta P_\ell(\cos\theta) \int_{-1}^{1} \frac{d(\cos\theta')}{2} \cos\theta' P_\ell(\cos\theta')$$

$$= \boxed{\frac{N(0)}{4} \sum_{\sigma\sigma'} \frac{F_{1\sigma\sigma'}}{9} v_F^2} \tag{46}$$

where we have used the fact that $\cos\chi = \cos\theta\cos\theta' + \sin\theta\sin\theta'\cos(\phi - \phi')$ and the integral of $\phi$ and $\phi'$ from 0 to $2\pi$ of $\cos(\phi - \phi')$ is zero.

(2)

$$2\sum_{pp'\sigma\sigma'} f_{pp'\sigma\sigma'} \cos\theta \cos\theta' \delta p_F \delta(p_F - p')\frac{\partial}{\partial p}\delta(p_F - p)$$

$$= -\frac{64\pi^2 p_F^3}{(2\pi\hbar)^6 v_F} \sum_{\ell\ell'\sigma\sigma'} \nu_{\ell\sigma} f_{\ell'\sigma\sigma'} \int_{-1}^{1} \frac{d(\cos\theta)}{2} P_\ell(\cos\theta) P_{\ell'}(\cos\theta) \cos\theta \int_{-1}^{1} \frac{d(\cos\theta')}{2} P_{\ell'}(\cos\theta') \cos\theta'$$

$$= -\frac{N(0)v_F}{p_F} \sum_{\ell\sigma\sigma'} \nu_{\ell\sigma} \frac{F_{1\sigma\sigma'}}{3} \begin{pmatrix} \ell & 1 & 1 \\ 0 & 0 & 0 \end{pmatrix}^2$$

$$= -\frac{N(0)v_F}{p_F} \sum_{\ell\sigma\sigma'} \nu_{\ell\sigma} \frac{F_{1\sigma\sigma'}}{3} \left\{ \frac{1}{3}\delta_{\ell0} + \frac{2}{15}\delta_{\ell2} \right\}$$

$$\approx \boxed{-\frac{N(0)}{4} \sum_{\ell\sigma\sigma'} \frac{4}{9} F_{1\sigma\sigma'} \frac{\nu_{\ell\sigma}}{m^*}\delta_{\ell0}} \tag{47}$$

where we have used the fact that

$$\begin{pmatrix} \ell & 1 & 1 \\ 0 & 0 & 0 \end{pmatrix} = (-1)^{\frac{\ell}{2}+1}\ell! \sqrt{\frac{(2-\ell)!}{(3+\ell)!}} \frac{\frac{\ell}{2}+1}{(1-\frac{\ell}{2})!\left(\frac{\ell}{2}\right)!} = -\frac{1}{\sqrt{3}}\delta_{\ell0} + \sqrt{\frac{2}{15}}\delta_{\ell2} \tag{48}$$

and we have assumed that we can safely ignore all terms dependent on $\ell > 1$.

(3)

$$\sum_{pp'\sigma\sigma'} f_{pp'\sigma\sigma'} \cos\theta \cos\theta' \delta p_F \delta p'_F \frac{\partial}{\partial p}\delta(p_F - p)\frac{\partial}{\partial p'}\delta(p_F - p')$$

$$\to \frac{1}{(2\pi\hbar)^6} \int d\Omega\, \delta p_F \cos\theta \int d\Omega'\, f_{pp'\sigma\sigma'}\, \delta p'_F \cos\theta' \int dp\, p^2 \frac{\partial}{\partial p}\delta(p_F - p) \int dp'\, p'^2 \frac{\partial}{\partial p'}\delta(p_F - p')$$

$$= \frac{N(0)}{p_F^2} \sum_{\ell\ell'\ell''\sigma\sigma'} \nu_{\ell\sigma}\nu_{\ell'\sigma'} F_{\ell''\sigma\sigma'} \begin{pmatrix} \ell'' & \ell & 1 \\ 0 & 0 & 0 \end{pmatrix}^2 \begin{pmatrix} \ell'' & \ell' & 1 \\ 0 & 0 & 0 \end{pmatrix}^2$$

$$\approx \frac{N(0)}{p_F^2} \sum_{\ell\ell'\ell''\sigma\sigma'} \nu_{\ell\sigma}\nu_{\ell'\sigma'} F_{\ell''\sigma\sigma'} \left\{ \frac{\delta_{\ell0}\delta_{\ell''1}}{3} + \frac{\delta_{\ell1}\delta_{\ell''0}}{3} \right\} \left\{ \frac{\delta_{\ell'0}\delta_{\ell''1}}{3} + \frac{\delta_{\ell'1}\delta_{\ell''0}}{3} \right\}$$

$$= \boxed{\frac{N(0)}{4} \sum_{\ell\sigma\sigma'} \nu_{\ell\sigma}\nu_{\ell\sigma'} \left\{ \frac{4F_{1\sigma\sigma'}}{3p_F^2}\delta_{\ell0} + \frac{4F_{0\sigma\sigma'}}{3p_F^2}\delta_{\ell1} \right\}} \tag{49}$$

where we have used the fact that

$$
\begin{pmatrix} \ell'' & \ell & 1 \\ 0 & 0 & 0 \end{pmatrix}^2 = \frac{(\ell - \ell'')^2 (1 + \ell + \ell'')}{(\ell + \ell'')(2 + \ell + \ell'')\Gamma(2 + \ell - \ell'')\Gamma(2 - \ell + \ell'')}
\tag{50}
$$

for $\ell'' + \ell$ odd and $\ell - 1 \le \ell'' \le 1 + \ell$. For $\ell = 0$, $\ell'' = 1$ while for $\ell = 1$, $\ell'' = 0$ or $\ell'' = 2$. Ignoring all terms higher than $\ell = 1$, we can then say that

$$
\begin{pmatrix} \ell'' & \ell & 1 \\ 0 & 0 & 0 \end{pmatrix}^2 \approx \frac{1}{3}\delta_{\ell 0}\delta_{\ell'' 1} + \frac{1}{3}\delta_{\ell 1}\delta_{\ell'' 0}
\tag{51}
$$

Bringing all terms that go as $A_\sigma \cdot A_{\sigma'}$ together, we find that it yields the following:

$$
\frac{N(0)}{4}\sum_{\sigma\sigma'} A_\sigma A_{\sigma'} \frac{F_{1\sigma\sigma'}}{9} v_F^2 - \frac{N(0)}{4}\sum_{\ell\sigma\sigma'} A_\sigma A_{\sigma'} \frac{4}{9} F_{1\sigma\sigma'} \frac{\nu_{\ell\sigma}}{m^*}\delta_{\ell 0}
$$
$$
+\frac{N(0)}{4}\sum_{\ell\sigma\sigma'} A_\sigma A_{\sigma'}\nu_{\ell\sigma}\nu_{\ell\sigma'}\left\{\frac{4F_{1\sigma\sigma'}}{3p_F^2}\delta_{\ell 0} + \frac{4F_{0\sigma\sigma'}}{3p_F^2}\delta_{\ell 1}\right\}
\tag{52}
$$

$$
=\frac{N(0)}{4}\frac{1}{3}\sum_{\sigma\sigma'} A_\sigma A_{\sigma'}\left[\frac{F_{1\sigma\sigma'}}{3}\left\{v_F^2 - 4\sum_\ell \frac{\nu_{\ell\sigma}}{m^*}\delta_{\ell 0}\right\} + \frac{4}{p_F^2}\sum_\ell \nu_{\ell\sigma}\nu_{\ell\sigma'}\{F_{1\sigma\sigma'}\delta_{\ell 0} + F_{0\sigma\sigma'}\delta_{\ell 1}\}\right]
\tag{53}
$$

We can then bring all the terms that go as $A_\sigma \cdot A_{\sigma'}$ together, and we simplify piece-by-piece:

$$
\sum_{\sigma\sigma'} A_\sigma A_{\sigma'}\frac{F_{1\sigma\sigma'}}{3}\left\{v_F^2 - 4\sum_\ell \frac{\nu_{\ell\sigma}}{m^*}\delta_{\ell 0}\right\}
$$
$$
= \frac{v_F^2}{3}\left(A_\uparrow^2(F_1^s + F_1^a) + 2A_\uparrow A_\downarrow(F_1^s - F_1^a) + A_\downarrow^2(F_1^s + F_1^a)\right)
$$
$$
-\frac{4}{m^*}\sum_\ell \delta_{\ell 0}\left(\nu_{\ell\uparrow}A_\uparrow^2(F_1^s + F_1^a) + \nu_{\ell\uparrow}A_\uparrow A_\downarrow(F_1^s - F_1^a) + \nu_{\ell\downarrow}A_\downarrow A_\uparrow(F_1^s - F_1^a) + \nu_{\ell\downarrow}A_\downarrow^2(F_1^s + F_1^a)\right)
$$

$$
= \begin{cases} \frac{4F_1^s}{3}A^2\left\{v_F^2 - \sum_\ell \frac{4\nu_{\ell s}}{m^*}\delta_{\ell 0}\right\}, & A_\uparrow = A_\downarrow \\ \frac{4F_1^a}{3}A^2\left\{v_F^2 - \sum_\ell \frac{4\nu_{\ell s}}{m^*}\delta_{\ell 0}\right\}, & A_\uparrow = -A_\downarrow \end{cases}
\tag{54}
$$

For the second term, we have

$$
\sum_{\ell\sigma\sigma'} A_\sigma A_{\sigma'}\frac{4}{p_F^2}\nu_{\ell\sigma}\nu_{\ell\sigma'}\{F_{1\sigma\sigma'}\delta_{\ell 0} + F_{0\sigma\sigma'}\delta_{\ell 1}\} \equiv \frac{4}{p_F^2}\sum_{\ell\sigma\sigma'} A_\sigma A_{\sigma'}\nu_{\ell\sigma}\nu_{\ell\sigma'}\widetilde{F}_{\ell\sigma\sigma'}
$$
$$
= \begin{cases} \frac{4A^2}{p_F^2}\sum_\ell \left(\widetilde{F}_\ell^s \nu_{\ell s}^2 + \widetilde{F}_\ell^a \nu_{\ell a}^2\right), & A_\uparrow = A_\downarrow \\ \frac{4A^2}{p_F^2}\sum_\ell \left(\widetilde{F}_\ell^s \nu_{\ell a}^2 + \widetilde{F}_\ell^a \nu_{\ell s}^2\right), & A_\uparrow = -A_\downarrow \end{cases}
\tag{55}
$$

We can then simplify the $A_\sigma \cdot A_{\sigma'}$ to the following:

$$\frac{N(0)}{4}\frac{1}{3}\sum_{\sigma\sigma'}A_\sigma A_{\sigma'}\left[\frac{F_{1\sigma\sigma'}}{3}\left\{v_F^2-4\frac{\nu_{\ell\sigma}}{m^*}\delta_{\ell0}\right\}+\frac{4}{p_F^2}\nu_{\ell\sigma}\nu_{\ell\sigma'}\left\{F_{1\sigma\sigma'}\delta_{\ell0}+F_{0\sigma\sigma'}\delta_{\ell1}\right\}\right]$$

$$=\frac{N(0)}{4}\left[\frac{4F_1^s}{9}v_F^2+\sum_\ell A^2\left\{\begin{array}{ll}-\frac{16F_1^s}{9m^*}\nu_{\ell s}\delta_{\ell0}+\frac{4}{3p_F^2}\left(\widetilde{F}_\ell^s\nu_{\ell s}^2+\widetilde{F}_\ell^a\nu_{\ell a}^2\right), & A_\uparrow=A_\downarrow \\ \\ -\frac{16F_1^a}{9m^*}\nu_{\ell s}\delta_{\ell0}+\frac{4}{3p_F^2}\left(\widetilde{F}_\ell^s\nu_{\ell a}^2+\widetilde{F}_\ell^a\nu_{\ell s}^2\right), & A_\uparrow=-A_\downarrow\end{array}\right.\right] \quad (56)$$

Let's try to simplify the total quadratic contribution to the free energy. We'll start with the case of $A_\uparrow=A_\downarrow$:

$$\frac{N(0)}{8}\left[\sum_\ell\left(\frac{1}{2\ell+1}\frac{1}{2\ell+1}\right.\right.$$
$$\times\left\{F_\ell^s\nu_{\ell s}^2+F_\ell^a\nu_{\ell a}^2+4A\left(v_F F_\ell^s\nu_{\ell s}\delta_{\ell0}-\frac{1}{p_F}\left\{\nu_{\ell s}^2\left[F_\ell^s+\frac{1}{2}(2\ell+1)F_0^s\right]+\nu_{\ell a}^2\left[F_\ell^a+\frac{1}{2}(2\ell+1)F_0^s\right]\right)\right)\right\}$$
$$+4A^2\left\{-\frac{2F_\ell^s}{m^*(2\ell+1)^2}\nu_{\ell s}\delta_{\ell0}+\frac{F_0^s}{p_F^2(2\ell+1)^2}(\nu_{\ell s}^2+\nu_{\ell a}^2)-\frac{16F_1^s}{9m^*}\nu_{\ell s}\delta_{\ell0}+\frac{4}{3p_F^2}\left(\widetilde{F}_\ell^s\nu_{\ell s}^2+\widetilde{F}_\ell^a\nu_{\ell a}^2\right)\right\}\right)+\left.\frac{4F_1^s}{9}v_F^2\right]$$

$$=\frac{N(0)}{8}\left[\sum_\ell\left(\frac{1}{2\ell+1}\frac{1}{2\ell+1}\right.\right.$$
$$\times\left\{F_\ell^s\nu_{\ell s}^2+F_\ell^a\nu_{\ell a}^2+\frac{4A}{p_F}\left(2\epsilon_F F_\ell^s\nu_{\ell s}\delta_{\ell0}-\left\{\nu_{\ell s}^2\left[F_\ell^s+\frac{1}{2}(2\ell+1)F_0^s\right]+\nu_{\ell a}^2\left[F_\ell^a+\frac{1}{2}(2\ell+1)F_0^s\right]\right\}\right)\right\}$$
$$+\frac{4A^2}{p_F^2}\left\{-4\epsilon_F\delta_{\ell0}\left[8F_1^s+\frac{F_\ell^s}{(2\ell+1)^2}\right]\nu_{\ell s}+\left[\frac{F_0^s}{(2\ell+1)^2}+\frac{4\widetilde{F}_\ell^s}{3}\right]\nu_{\ell s}^2+\left[\frac{F_0^s}{(2\ell+1)^2}+\frac{4\widetilde{F}_\ell^a}{3}\right]\nu_{\ell a}^2\right\}\right)+\left.\frac{64A^2}{9p_F^2}F_1^s\epsilon_F^2\right]$$

$$\approx\boxed{\frac{N(0)}{8}\left[\sum_\ell\frac{1}{2\ell+1}\left(\frac{1}{2\ell+1}\left\{F_\ell^s\nu_{\ell s}^2+F_\ell^a\nu_{\ell a}^2\right\}+\frac{8A}{p_F}\frac{F_\ell^s}{2\ell+1}\epsilon_F\nu_{\ell s}\delta_{\ell0}\right)+\frac{64A^2}{9p_F^2}F_1^s\epsilon_F^2\right]} \quad (57)$$

We now do the same simplification for the spin gauge $A_\uparrow=-A_\downarrow$:

$$\frac{N(0)}{8}v\sum_\ell\left(\frac{1}{2\ell+1}\frac{1}{2\ell+1}\left\{F_\ell^s\nu_{\ell s}^2+F_\ell^a\nu_{\ell a}^2+4A\left(v_F F_\ell^s\nu_{\ell a}\delta_{\ell0}-\frac{1}{p_F}\nu_{\ell s}\nu_{\ell a}\left\{F_\ell^s+F_\ell^a+(2\ell+1)F_0^s\right\}\right)\right\}\right.$$
$$+4A^2\left\{-\frac{2F_\ell^s}{m^*(2\ell+1)^2}\nu_{\ell s}\delta_{\ell0}+\frac{F_0^s}{p_F^2(2\ell+1)^2}(\nu_{\ell s}^2+\nu_{\ell a}^2)-\frac{16F_1^a}{m^*}\nu_{\ell s}\delta_{\ell0}+\frac{4}{3p_F^2}\left(\widetilde{F}_\ell^s\nu_{\ell a}^2+\widetilde{F}_\ell^a\nu_{\ell s}^2\right)\right\}\right)+\left.\frac{4F_1^a}{9}v_F^2\right]$$

$$=\frac{N(0)}{8}\left[\sum_\ell\left(\frac{1}{2\ell+1}\frac{1}{2\ell+1}\left\{F_\ell^s\nu_{\ell s}^2+F_\ell^a\nu_{\ell a}^2+\frac{4A}{p_F}\nu_{\ell a}\left(2\epsilon_F F_\ell^s\delta_{\ell0}-\nu_{\ell s}\left\{F_\ell^s+F_\ell^a+(2\ell+1)F_0^s\right\}\right)\right\}\right.\right.$$
$$+\frac{4A^2}{p_F^2}\left\{-4\epsilon_F\frac{F_\ell^s}{(2\ell+1)^2}\nu_{\ell s}\delta_{\ell0}+\frac{F_0^s}{(2\ell+1)^2}(\nu_{\ell s}^2+\nu_{\ell a}^2)-\frac{32}{9}\epsilon_F^2F_1^a\nu_{\ell s}\delta_{\ell0}+\frac{4}{3}\left(\widetilde{F}_\ell^s\nu_{\ell a}^2+\widetilde{F}_\ell^a\nu_{\ell s}^2\right)\right\}\right)+\left.\frac{64A^2}{9p_F^2}F_1^a\epsilon_F^2\right]$$

$$\approx\boxed{\frac{N(0)}{8}\left[\sum_\ell\frac{1}{2\ell+1}\left(\frac{1}{2\ell+1}\left\{F_\ell^s\nu_{\ell s}^2+F_\ell^a\nu_{\ell a}^2\right\}+\frac{8A}{p_F}\frac{F_\ell^s}{2\ell+1}\epsilon_F\nu_{\ell a}\delta_{\ell0}\right)+\frac{64A^2}{9p_F^2}F_1^a\epsilon_F^2\right]}$$

$$(58)$$

Putting this altogether, we find that

$$\sum_{pp'\sigma\sigma'} f_{pp'\sigma\sigma'} \delta n_{p-A_\sigma,\sigma} \delta n_{p'-A_{\sigma'},\sigma'}$$

$$= \begin{cases} \frac{N(0)}{8} \left[ \sum_\ell \frac{1}{2\ell+1} \left( \frac{1}{2\ell+1} \left\{ F_\ell^s \nu_{\ell s}^2 + F_\ell^a \nu_{\ell a}^2 \right\} + \frac{8A}{p_F} \frac{F_\ell^s}{2\ell+1} \epsilon_F \nu_{\ell s} \delta_{\ell 0} \right) + \frac{64A^2}{9p_F^2} F_1^s \epsilon_F^2 \right], & A_\uparrow = A_\downarrow \\[4mm] \frac{N(0)}{8} \left[ \sum_\ell \frac{1}{2\ell+1} \left( \frac{1}{2\ell+1} \left\{ F_\ell^s \nu_{\ell s}^2 + F_\ell^a \nu_{\ell a}^2 \right\} + \frac{8A}{p_F} \frac{F_\ell^s}{2\ell+1} \epsilon_F \nu_{\ell a} \delta_{\ell 0} \right) + \frac{64A^2}{9p_F^2} F_1^a \epsilon_F^2 \right] & A_\uparrow = -A_\downarrow \end{cases}$$

This completes the calculation of Eqn. (5).