# Peer review of "Gauging away the Stoner model: Engineering unconventional metallic ferromagnetism with artificial gauge fields"

_SciPost Physics Core_

## Round 1 · Referee Report · Anonymous (Referee 1) · 2020-12-16

Report

The work of Heath and Bedell deals with the interesting topic of Landau theory in the presence of gauge potentials, and explores their effect on the Stoner instability to ferromagnetism. The article might be relevant for a range of experiments on correlated fermionic systems and, in particular, for ultracold gases. However, it is not clear to me what is the concrete microsopic model underlying the Landau theory described here.

Of course, the vector potential should be included in the Hamiltonian via the minimal coupling \vec{p}->\vec{p}-\vec{A}. Such potential in general has a non-trivial coordinate dependence, which does not seem to correspond to the present case, because translational invariance is not broken (the momentum p remains a good quantum number for quasiparticle excitations throughout the article).

A simpler case where quasiparticles with well-defined p are naturally defined occurs if the gauge potential is constant in space, as it might be suggested by the remark at page 3: "For some weak non-zero gauge field, the momentum of every particle in the Fermi sea is shifted by the same amount". But calculations seem to contradict this interpretation, because the Fermi surface is not shifted rigidly (\vec{p}_F->\vec{p}_F-\vec{A}, with \vec{A} a constant vector), but has a modified Fermi radius - see for example Eqs. 3 and 4. I do not see from the article how such a p-dependent (but constant in magnitude) vector potential could arise concretely. Note that here the potential always commutes with sigma_z, which is different, e.g. from a pure Rashba spin-orbit interaction (A_x~A \sigma_y and A_y~-A sigma_x, leading to modified Fermi radii). Therefore, I strongly suggest the Author to clearly explain the physical setting they have in mind, even at the risk of stating the obvious.

Other more specific suggestions are the following:

  • In the introduction, the Authors emphasize the observation of spin segregation as a main motivation of their study. First, I would have appreciated in the beginning a short passage which defines spin segregation and explain the phenomenology the Authors have in mind (contrasting it explicitly with ferromagnetism). Furthermore, I missed in the main body of the article a discussion on how their theory could lead to segregation together or without ferromagnetism, in different regimes of the Laundau theory.

  • A related question is on how the present theory could be applied concretely to specific cases. For example, for the Li ultracold gases mentioned repeatedly in the manuscript, the Authors could review the gauge potential entering that particular system, and discuss how it translates to the Laudau theory given here. This would make the theory more useful and also help to clarify my previous question on the gauge potential.

-In Eqs. 4 and 5, I am confused by the subscript "s/a" in the summation. nu_l,s and nu_l,a are different quantities, defined just below Eq. 4, so I am not sure which ones of the two enters the summation, or which of the two cases the s/a refers to (I do not see any alternatives on the left-hand-side).

Below Eq. 6, one of the two quantities is always taken to be zero, but here the summation also involves vu_ls^2+nu_la^2. So it seems that both quantities are generally non-zero in Eq. 4 (otherwise the notation nu_l,a/s^2 would apply?). I suggest to clarify the formula.

  • Eq. 6 is rather puzzling to me, because I would have expected that when A->0 the standard instability conditions are recovered. However, the conditions given here do not contain A and the limit cannot be taken in a straightforward way. The connection with the standard Stoner instability at A=0 seems rather subtle from this result, and should be probably discussed explicitly.

  • Figure 1 is rather unclear to me. Why the y axis is labeled with a quantity which is a function of F_0^s (the x-axis)? Usually the axes refer to independent physical quantities. It seems that the y-axes F_1^s and the blue curves are given by Eq. 6a (referred to in the caption)? However, Eq. 6a states that the the stability condition is derived under the assumption F_0^s<-1, but in the figure also F_0^s>-1 is included. Figure 2 is even more complicated and in my opinion suffers of similar problems.

In conclusion, the article might contain interesting result but it is difficult for me to judge their actual validity and significance from the present manuscript. Therefore, I suggest to reject the current version of the article.

---

## Round 1 · Referee Report · Anonymous (Referee 2) · 2021-1-6

Report

The manuscript by Heath and Bedell discusses the stability of a Fermi liquid in presence of an external gauge field. The math presented in the Supplemental Material is possibly correct, but the main article is in in my opinion very obscure under many aspects. A key point which is left unclear for example is how do the authors define "spin segregation" and "ferromagnetic ordering". It is also unclear how the present calculation recovers the standard results in absence of external field (as $A\rightarrow0$ ). Moreover, the graphs presented are very difficult to understand. Generally, I have the impression that the authors used exceedingly difficult phrases without need. In conclusion, I do not recommend publication of the present manuscript in SciPost. Here below follow some further more specific remarks and comments, listed in chronological order, which the authors may want to take into account.

Many quantities are poorly defined. Some (like $p$ and $\sigma$) are relatively obvious. But for example, how many spin states do the authors consider is left unspecified. A "spin 1/2 Fermi gas" is mentioned in the Intro, and I'm guessing this is the chosen system. Maybe for clarity the authors should have restated that this is the case of interest right before Eq.(1)? Then, two lines below Eq.(2) variables like $n_p$ and $\nu_p$ have a single index, but they had two indices when first introduced. And what is $k$ in this relation? Is it the same as $p$, or something else? And $\nu$ has no indices whatsoever in Eq. (9).

Later on, it is also unclear what is the definition of η (the dimensionless phase velocity of zero sound): how is that made adimensional?

Similarly, the "Pomeranchuk condition" is mentioned, but undefined. Is this $1+F<0$?

The sentences " In the limit of weak (i.e., local) ferromagnetism, .... It therefore becomes apparent that ... higher non-vanishing orbital partial waves [37]." are very obscure, and should be duly clarified.

In the sentence "the momentum of every particle in the Fermi sea is shifted by the same amount: $p_F → p_F − A_\sigma$, I think $p_F$ should be replaced by $p$.

If the second term in Eq.(5) remains only finite for the $l=0$ channel, the expression would be much more readable if $l$ was replaced by $0$ (except obviously in the delta function $\delta_{l0}$; i.e., $f_l\delta_{l0}=f_0\delta_{l0}$).

The same applies for various other equations, like for example (25), (27), (28), ...

The discussion on the "condensation of gauge bosons" is exceedingly unclear. What do the authors refer to with "the fundamental carriers of the gauge field"? And why do the mentioned conditions imply their "Bose condensation"? And what do the authors mean by "nonintegrable phase attached to the electron field operators"?

The graphs in Figs. 1 and 2 are very obscure. What is the physical meaning of $\nu_{0a}=0$, or $\nu_{0s}=0$? And what happens when both $\nu_{0a}$ and $\nu_{0s}$ are non-zero?

Eq. (9) appears out of nowhere. For completeness, I think the Authors should have at least introduced explicitly Landau's kinetic equation, and briefly outlined how Eq. (9) arises out of it.

$\theta_0$ appearing in Eqs. (10a) and (10b) is, as far as I noticed, undefined.

The matrix-shaped objects appearing in Eqs. (34) and (35), arising from the integration over products of Legendre polynomials, are also undefined. Are those Wigner matrices?

---

## Editorial Decision

awaiting_resubmission